# Taxonomic Reassessment of the Izumo Lineage of *Hynobius utsunomiyaorum*: Description of a New Species from Chugoku, Japan

**DOI:** 10.3390/ani11082187

**Published:** 2021-07-23

**Authors:** Hirotaka Sugawara, Takayuki Iwata, Hitoshi Yamashita, Masahiro Nagano

**Affiliations:** 1Faculty of Science and Technology, Kochi University, 2-5-1 Akebonocho, Kochi City 780-8072, Japan; chiropterotriton@yahoo.co.jp; 2Yasugi City Hall, 878-2, Yasugicho, Yasugi City 692-8686, Japan; urodela.anura@gmail.com; 3Kurayoshi City Hall, 722, Aoimachi, Kurayoshi City 682-8611, Japan; outdoor.60.12.21@gmail.com; 4Faculty of Science and Technology, Oita University, 700 Dannoharu, Oita City 870-1192, Japan

**Keywords:** canonical discriminant analysis, cytochrome *b*, lentic salamander, polyphyly, San’in region

## Abstract

**Simple Summary:**

In the study, we investigated the true taxonomic composition of *Hynobius utsunomiyaorum* from Chugoku District, Japan. Our detailed morphological and molecular comparisons showed that *H. utsunomiyaorum* is clearly divided into two species: the true *H. utsunomiyaorum* and the Izumo Lineage of *H. utsunomiyaorum*, i.e., *Hynobius kunibiki* sp. nov. This new species does not satisfy the diagnosis of *H. utsunomiyaorum*, while these two species do not form a monophyletic group based on molecular data. *Hynobius setoi* is morphologically more similar to the new species than is *H. utsunomiyaorum*. Previous studies have suggested that *H. setoi* is distributed across the San’in region, i.e., the northern part of Chugoku District, Japan. However, our research revealed that *H. setoi* is not distributed in the northern part of Shimane Prefecture, located in the western part of the San’in region in Chugoku District. Rather than *H. setoi*, we found that the new species was distributed in the northern part of Shimane Prefecture. Additionally, results of phylogenetic analyses using all valid Japanese *Hynobius* species are provided in our study. Based on these results, we show that Japanese *Hynobius* species included in the subgenus *Hynobius* may be divided into four genetic clades. The information in our study will be vital for developing conservation management strategies and policies for these species.

**Abstract:**

Here, we describe a new species of the genus *Hynobius* from Chugoku, Japan. In populations from central to eastern Shimane Prefecture, the Izumo Lineage of *Hynobius utsunomiyaorum* was clearly distinguished from the true *Hynobius utsunomiyaorum* based on morphological and molecular evidence. Thus, we describe the former lineage as a new species, *Hynobius kunibiki* sp. nov. Morphological comparisons revealed that *H. utsunomiyaorum* lacks a distinct yellow line on the ventral side of its tail, whereas the new species possesses this yellow line; most *H. utsunomiyaorum* individuals have distinct white spots on the lateral sides of their body and lack a fifth toe, whereas the new species largely lacks these spots, and all examined individuals had a fifth toe. The two species also differed significantly by several other morphological characteristics. The lentic species *Hynobius setoi* is morphologically similar to *H. kunibiki* sp. nov., but they differ significantly by various morphological characteristics. Despite their partial morphological similarity, these two species differed substantially in terms of their genetics. Finally, we show, in a phylogenetic tree including all Japanese *Hynobius* species, that the subgenus *Hynobius* can be divided into four genetic clades. Overall, this information will help develop conservation management strategies and policies for these species.

## 1. Introduction

In the family Hynobiidae (the Asiatic salamanders), the most diverse genus *Hynobius* includes 55 species, 37 (67.3%) of which are endemic to Japan [1]. The identification of *Hynobius* species can be challenging without locality and genetic data because they have few distinguishing morphological characteristics [2]. Previous studies have revealed that cryptic species of lotic or lentic *Hynobius* exist in Japan [3,4]; that are yet to be described [3,5].

The Hiba salamander, *Hynobius utsunomiyaorum*, was described from Rokunohara, Saijocho Yuki, Shobara City, Hiroshima Prefecture, and has been found in the Chugoku Mountains in eastern Chugoku District, Japan [3]. According to Matsui et al. [3], this species is not distributed in the northern part of Shimane Prefecture; instead, the typical pond breeding species, *Hynobius setoi*, is found in this area [3]. However, genetic evidence that *H. setoi* is distributed in northern Shimane Prefecture was not provided by Matsui et al. [3]; such evidence came only from morphological data. On the other hand, Hayashi and Ooi [5] sampled widely from the northern part of Shimane Prefecture and then performed mitochondrial DNA analyses on these samples. Results showed that haplotypes of *H. utsunomiyaorum* but not *H. setoi* were found in the study area [5]. However, populations from the northern part of Shimane Prefecture are clearly distinguishable from true *H. utsunomiyaorum* in terms of their morphology and ecology; thus, Hayashi and Ooi [5] named these populations the Izumo Lineage of *H. utsunomiyaorum* and suggested that they may be a new species. They concluded that *H. setoi* is not distributed in the northern part of Shimane Prefecture [5]; however, detailed morphological evidence for these populations is currently lacking.

The Izumo Lineage suggested by Hayashi and Ooi [5] corresponds to the B2b3 clade suggested by Matsui et al. [3]. Based on mitochondrial DNA and allozyme analyses, this lineage paraphyletic with respect to *H. utsunomiyaorum* [3,6]; therefore, based on the phylogenetic species concept, it should be considered a new species. When a new species is described, essential features must be directly derived from characteristics observed in specimens rather than indirectly inferred [7]; hence, the morphological characteristics that distinguish a species from its close relatives should be described in a “diagnosis.″ If the Izumo Lineage does not satisfy the diagnosis conditions of *H. utsunomiyaorum*, the two should be considered different species based on the morphological species concept. In herpetology, Frost and Hillis [8] recommend the use of the phylogenetic and evolutionary species concepts. If the Izumo Lineage does not form a monophyly with *H. utsunomiyaorum* after DNA analysis of samples from the entire distribution range of *H. utsunomiyaorum*, the two should be considered different species based on the phylogenetic and evolutionary species concepts. Although Matsui et al. [3] concluded that *H. setoi* is distributed in the northern part of Shimane Prefecture based on its morphology, the phylogenetic analysis of Hayashi and Ooi [5] suggests that *H. setoi* from the northern part of Shimane Prefecture was most likely mistaken with the Izumo Lineage because the two share a similar morphology. Thus, morphological comparisons between *H. setoi* and the Izumo Lineage of *H. utsunomiyaorum* are also required.

In the present study, we used additional DNA sequence data to reconstruct the phylogeny of *Hynobius* collected from Chugoku. Furthermore, we statistically analyzed morphological characteristics to compare two species and one candidate species, i.e., *H. utsunomiyaorum*, the Izumo Lineage of *H. utsunomiyaorum*, and *H. setoi*. We also evaluated the species validity of the Izumo Lineage of *H. utsunomiyaorum* using the phylogenetic, evolutionary, and morphological species concepts. We revealed the detailed distribution ranges and borders of *H. utsunomiyaorum*, the Izumo Lineage of *H. utsunomiyaorum*, and *H. setoi* by the sampling from their complete distribution ranges; additionally, we investigated the distribution ranges of *Hynobius iwami* in the eastern part of Shimane Prefecture, which are found near the distribution ranges of the Izumo Lineage of *H. utsunomiyaorum*. Finally, we constructed a phylogenetic tree including all Japanese *Hynobius* species and estimated the phylogenetic relationships in this genus.

## 2. Materials and Methods

### 2.1. Molecular Analysis

For phylogenetic analysis, we sampled 70 individuals (from February 2007 to January 2021) representing the Izumo Lineage of *H.*
*utsunomiyaorum* (27 individuals), *H. setoi* (24 individuals), *H. utsunomiyaorum* (13 individuals), *H. iwami* (three individuals), *Hynobius abuensis* (one individual from the type locality), *Hynobius*
*tsurugiensis* (one individual from the type locality), *Hynobius*
*kuishiensis* (one individual from the type locality), and *Hynobius*
*guttatus* (one individual not from the type locality) from 70 localities (Appendix A; Figure 1). For samples taken during the breeding season, we removed a single tailbud embryo from each paired egg sac and preserved it in 99.5% ethanol. We also obtained tissue samples by clipping the caudal extremities from larvae preserved in 99.5% ethanol. Compared with other groups of Japanese *Hynobius*, it is more difficult to collect eggs and larvae from the *Hynobius*
*stejnegeri* group, including *H. tsurugiensis*, *H. kuishiensis*, and *H. guttatus*; therefore, we obtained tissue samples from the caudal extremities of adults in these three species. We extracted total genomic DNA from the tail tips or single tailbud embryos and preserved it in 99.5% ethanol using a DNeasy Blood and Tissue Kit (Qiagen, Hilden, Germany). For all individual *Hynobius*, we amplified a 630-bp fragment of the cytochrome *b* gene using Ex Taq^®^ (TaKaRa, Tokyo, Japan) with primers L14010 (5′-TAHGGWGAHGGATTWGAWGCMACWGC-3′) and H14778 (5′-AARTAYGGGTGRAADGRRAYTTTRTCT-3′) [9]. The reaction mix (with a total volume of 10 μL) contained 1.0 μL of 10× Ex Taq Buffer, 0.8 μL of 25-mM dNTP mix, 0.5 μL of each of the forward and reverse primers (10 pM), 0.05 μL of Taq polymerase, 6.15 μL of distilled deionized water, and 1.0 μL of template DNA. A T100™ thermal cycler (Bio-Rad, Hercules, CA, USA), was utilized with the following conditions: an initial 3-min denaturing step at 94 °C; 40 cycles of 30 s at 94 °C, 45 s at 56 °C, and 90 s at 72 °C; with a final 10-min extension at 72 °C. We purified PCR products with Illustra™ ExoStar™ 1-Step (GE Healthcare, Buckinghamshire, UK) and sequenced them using BigDye^®^ Terminator ver. 3.1 (Applied Biosystems, Foster City, CA, USA) on an ABI 3130xl Genetic Analyzer (Applied Biosystems). We deposited the acquired sequences into the DNA Data Bank of Japan (Appendix A). We aligned the DNA sequences using MEGA X [10], and then performed phylogenetic analyses of the aligned sequences from all recognized Japanese *Hynobius* species, with “*Salamandrella keyserlingii*″ used as the outgroup (Appendix A), using Bayesian inferences (BIs) and maximum likelihood (ML) estimates. We estimated the best-fit nucleotide substitution model using jModelTest 2 [11] based on the corrected Akaike’s information criterion [12] and Bayesian information criterion [13]. We selected the Hasegawa–Kishino–Yano model (gamma distribution with invariant sites) based on both criteria. We constructed Bayesian and ML trees using MrBayes 3.2 [14] and MEGA X [10], respectively. For the Bayesian analysis, we performed two independent MCMC runs for 5,000,000 generations with a sample frequency of 100. We assessed the statistical support for monophyly with posterior probabilities and examined the stationarity of the likelihood scores of sampled trees using Tracer version 1.7 (http://tree.bio.ed.ac.uk/software/tracer/) (accessed on 17 February 2021). The first 25% of generations were discarded as burn-in. We regarded a group as monophyletic when a node had a posterior probability ≥0.95 according to the criterion of Huelsenbeck and Rannala [15]. For ML analysis, we estimated the statistical support for monophyly with 1000 bootstrap replicates; a group was regarded as monophyletic when a node had a bootstrap value ≥80% according to the criterion of Regier et al. [16].

### 2.2. Morphological Analysis

We sampled 124 individuals across the entire distribution areas of three groups from January 2019 to March 2021, including 31 individuals of the Izumo Lineage of *H. utsunomiyaorum* (22 males and nine female) from seven populations (Pops. 1–3, 8, 17, 22, and 23), 45 individuals of *H. setoi* (30 males and 15 females) from six populations (Pops. 75, 76, 81, 82, 84, and 87), and 48 individuals of *H. utsunomiyaorum* (38 males and 10 females) from four populations (Pops. 100, 102, 111, and 112) (Appendix A; Figure 1).

From a conservation perspective, the collected specimens were measured under anesthesia and subsequently returned to their site of capture except for the candidate individuals of type specimens (holotype and two paratypes of both sexes). Before specimens were returned, we captured images of the dorsal, ventral, and lateral sides of all individuals on a black background. We also obtained tissue samples (preserved in 99.9% ethanol) from the tail tips of all individuals as evidence that they had been collected. Salamanders were anesthetized with ethyl 3-aminobenzoate methanesulfonate salt (Sigma-Aldrich, St. Louis, MO, USA) diluted 1000-fold with water [17]. Using a digital caliper, 22 measurements were made to the nearest 0.1 mm on anesthetized specimens, including the following: snout–vent length (SVL), trunk length (TRL), axilla–groin distance (AGD), head length (HL), tail length (TAL), median tail width (MTAW), median tail height (MTAH), vomerine teeth length (VTL), and vomerine teeth width (VTW), head width (HW), forelimb length (FLL), hindlimb length (HLL), second finger length (2FL), third finger length (3FL), third toe length (3TL), fifth toe length (5TL), internarial distance (IND), interorbital distance (IOD), upper eyelid length, snout length (SL), upper eyelid width, and lower jaw length (LJL). For each individual, we also recorded data on the presence and number of distinct white spots on the venter (DWSV) and lateral (DWSL) sides of the body, the presence of a distinct and continuous yellow line on the dorsal side of the tail (DCTYLD), the presence of a distinct yellow line on the ventral side of the tail (DTYLV), and the presence of distinct gular mottling (DGM). The number of costal folds between the adpressed limbs (CFBALN) and the number of costal grooves (CGN) were counted; we used the counting method of Matsui et al. [3] to count CGN.

Prior to performing morphological comparisons among the three groups, we tested for normality using a Shapiro–Wilk test. When data were found to follow a normal distribution, we tested for homoscedasticity using Bartlett’s test. When the variances among the populations were found to be equal, we performed Tukey–Kramer tests; when variances were found not to be equal, we performed Games–Howel tests. When data did not follow a normal distribution and variances among populations were not equal, we performed Steel–Dwass tests. To examine the overall morphological variation among the three groups, we performed canonical discriminant analysis using standardized values for the 22 measurements. We also performed morphological comparisons between males and females for the three groups. Before the analyses, we tested for normality using a Shapiro–Wilk test. When data followed a normal distribution, we tested for homoscedasticity using an F test. When the variances among the groups were equal, we performed Student’s *t* tests; when variances were not equal, we performed Welch’s *t* test. When data did not follow a normal distribution and variances among groups were not equal, we performed Brunner–Munzel tests. All statistical analyses were performed in R with α = 0.05 as the significance level [18]. Additionally, we used R to calculate the Euclidean distances among the three groups based on the 22 morphological characters in both sexes.

### 2.3. Sampling and Measurements of Type Specimens

We collected a holotype and one paratype female from Pop. 1 on 20 December 2019 (Nishinomura, Kamionocho, Matsue City, Shimane Prefecture). We also collected one paratype male from Pop. 17 (Kamocho Chikamatsu, Unnan City, Shimane Prefecture) on 20 January 2021. The collected type specimens were fixed in 10% formalin and then transferred to 70% ethanol. The preserved holotype specimens (TRPM-ARA-0000014) were deposited in Tottori Prefectural Museum (2-124, Higashimachi, Tottori City, Tottori Prefecture, 680-0011, Japan). The two preserved paratype specimens were deposited in Yokosuka City Museum (95, Fukadadai, Yokosuka City, Kanagawa Prefecture, 238-0016, Japan; paratype female: YCM-RA589; paratype male: YCM-RA590). We took 43 measurements from the holotype including SVL, TRL, AGD, HL, TAL, MTAW, MTAH, basal tail width (BTAW), basal tail height (BTAH), VTL, VTW, HW, maximum head width (MXHW), left forelimb length (LFLL), left hindlimb length (LHLL), right forelimb length (RFLL), right hindlimb length (RHLL), left first finger length (L1FL), left second finger length (L2FL), left third finger length (L3FL), left fourth finger length (L4FL), right first finger length (R1FL), right second finger length (R2FL), right third finger length (R3FL), right fourth finger length (R4FL), left first toe length (L1TL), left second toe length (L2TL), left third toe length (L3TL), left fourth toe length (L4TL), left fifth toe length (L5TL), right first toe length (R1TL), right second toe length (R2TL), right third toe length (R3TL), right fourth toe length (R4TL), right fifth toe length (R5TL), IND, IOD, left upper eyelid length (LUEL), right upper eyelid length (RUEL), SL, left upper eyelid width (LUEW), right upper eyelid width (RUEW), and LJL. We also counted the CGN using the method of Matsui et al. [3].

## 3. Results

### 3.1. Molecular Analysis

Phylogenetic reconstructions using BI and ML with the cytochrome *b* gene recovered nearly identical trees. The Izumo Lineage of *H.*
*utsunomiyaorum* (Pops. 1–71) and *H. utsunomiyaorum* (Pops. 100–131) did not form a monophyletic group, whereas the former group and *Hynobius akiensis* do (Figure 2). *Hynobius setoi* was largely separated from the two groups (Figure 2); this species formed a monophyletic group which includes *Hynobius mikawaensis*, *Hynobius takedai*, and *Hynobius nigrescens* (Figure 2). Haplotypes of *H. setoi* were found from Pop. 72 (Tai, Shinonsen Town, Hyogo Prefecture) to Pop. 95 (Higashiizumocho Iya, Matsue City, Shimane Prefecture) (Figure 1 and Figure 2) but were not detected from the northern central part of Shimane Prefecture. In contrast, haplotypes of the Izumo Lineage of *H. utsunomiyaorum* were found from the central to northern part of Shimane Prefecture (Figure 1 and Figure 2). Indeed, the Izumo Lineage was found from Pop. 14 (Hirosecho Hirose, Yasugi City, Shimane Prefecture) to Pop. 27 (Asayamacho Senyama, Oda City, Shimane Prefecture) (Figure 1 and Figure 2). In addition, haplotypes of *H. utsunomiyaorum* were detected from Pop. 106 (Nagatani, Iinan Town, Shimane Prefecture) to Pop. 112 (Ogaya, Nishiawakura Village, Okayama Prefecture) (Figure 1 and Figure 2). Finally, haplotypes of *H. iwami* were newly discovered from Pop. 132 (Asayamacho Asakura, Oda City, Shimane Prefecture) to Pop. 134 (Oda, Sakuraecho, Gotsu City, Shimane Prefecture) (Figure 1 and Figure 2).

### 3.2. Morphological Analysis

Morphological measurements of the three groups are shown in Table 1. Males of the Izumo Lineage of *H. utsunomiyaorum* and *H. setoi* differed significantly in seven morphological characteristics: RTRL (*p* < 0.05), RHL (*p* < 0.05), RTAL (*p* < 0.05), RIND (*p* < 0.001), RIOD (*p* < 0.01), RSL (*p* < 0.0001), and RUEL (*p* < 0.001). Additionally, females of the Izumo Lineage of *H. utsunomiyaorum* and *H. setoi* differed significantly in four morphological characteristics: RVTL (*p* < 0.01), RFLL (*p* < 0.05), RIOD (*p* < 0.05), and RSL (*p* < 0.01). Furthermore, significant differences were detected between the males of the Izumo Lineage of *H. utsunomiyaorum* and *H. utsunomiyaorum* for seven morphological characteristics: SVL (*p* < 0.0001), RTAL (*p* < 0.0001), RMTAH (*p* < 0.0001), RVTL (*p* < 0.0001), R3FL (*p* < 0.01), R5TL (*p* < 0.0001), and RSL (*p* < 0.01). Significant differences were also detected between the females of the Izumo Lineage of *H. utsunomiyaorum* and *H. utsunomiyaorum* for four morphological characteristics: RAGD (*p* < 0.05), RMTAH (*p* < 0.05), R3TL (*p* < 0.05), and R5TL (*p* < 0.01). Moreover, significant differences existed between the males of *H. setoi* and *H. utsunomiyaorum* in 15 morphological characteristics: SVL (*p* < 0.0001), RTRL (*p* < 0.01), RHL (*p* < 0.001), RTAL (*p* < 0.05), RMTAH (*p* < 0.0001), RVTL (*p* < 0.01), RVTW (*p* < 0.05), R3FL (*p* < 0.0001), R5TL (*p* < 0.0001), RIND (*p* < 0.001), RIOD (*p* < 0.01), RUEW (*p* < 0.05), RSL (*p* < 0.001), RUEL (*p* < 0.01), and RLJL (*p* < 0.001). Similarly, significant differences were found between the females of *H. setoi* and *H. utsunomiyaorum* in 12 morphological characteristics: SVL (*p* < 0.01), RAGD (*p* < 0.05), RHL (*p* < 0.01), RMTAH (*p* < 0.001), RVTW (*p* < 0.05), RFLL (*p* < 0.05), RHLL (*p* < 0.01), R2FL (*p* < 0.05), R3TL (*p* < 0.01), R5TL (*p* < 0.0001), RIOD (*p* < 0.05), and RSL (*p* < 0.05). Canonical discriminant analyses indicated that the three groups differed according to their males and females (Figure 3). Euclidean distances among the three groups were as follows: Izumo Lineage of *H. utsunomiyaorum* vs. *H. setoi* = 4.76 in males and 6.00 in females; Izumo Lineage of *H. utsunomiyaorum* vs. *H. utsunomiyaorum* = 11.27 in males and 5.96 in females; and *H. setoi* vs. *H. utsunomiyaorum* = 9.84 in males and 10.27 in females.

Results of morphological observations are summarized in Table 2. Males of the Izumo Lineage of *H. utsunomiyaorum* mostly lacked distinct white spots on the ventral (21/22 = 95.5%) and lateral (21/22 = 95.5%) sides of the body, but they largely possessed a distinct and continuous yellow stripe on the dorsal (22/22 = 100%) side of the tail, a distinct yellow stripe on the ventral (22/22 = 100%) edge of the tail, DGM (18/22 = 81.8%), and 12 costal grooves (18/22 = 81.8%). Females of the Izumo Lineage of *H. utsunomiyaorum* lacked distinct white spots on the lateral (9/9 = 100%) sides of the body as well as DGM (9/9 = 100%), but they possessed a distinct and continuous yellow stripe on the dorsal (9/9 = 100%) side of the tail and a distinct yellow stripe on the ventral (9/9 = 100%) edge of the tail. Males of *H. setoi* possessed a distinct and continuous yellow stripe on the dorsal (30/30 = 100%) side of the tail, a distinct yellow stripe on the ventral (30/30 = 100%) edge of the tail, and mostly had 12 costal grooves (24/30 = 80.0%); however, they largely lacked distinct white spots on the ventral (21/30 = 70.0%) and lateral (28/30 = 93.3%) sides of the body. Females of *H. setoi* possessed a distinct and continuous yellow stripe on the dorsal side of the tail (15/15 = 100%), a distinct yellow stripe on the ventral (15/15 = 100%) edge of the tail, but they lacked DGM (15/15 = 100%), and mostly lacked distinct white spots on the lateral sides of the body (13/15 = 86.7%). Males of *H. utsunomiyaorum* lacked a distinct yellow stripe on the ventral side of the tail (38/38 = 100%) and mostly lacked a distinct and continuous yellow stripe on the dorsal side of the tail (33/38 = 86.8%) as well as DGM (29/38 = 76.3%); however, they largely possessed distinct white spots on the ventral (30/38 = 78.9%) and lateral (34/38 = 89.5%) sides of the body. Females of *H. utsunomiyaorum* lacked a distinct yellow stripe on the ventral side of the tail (10/10 = 100%) along with DGM (10/10 = 100%), but they mostly had distinct white spots on the ventral (9/10 = 90.0%) and lateral (9/10 = 90.0%) sides of the body as well as 12 costal grooves (8/10 = 80.0%).

From the perspectives of the three species concepts, we can describe the Izumo Lineage of *H. utsunomiyaorum* as a new species based on our morphological and molecular analyses.

## 4. Species Account

*Hynobius kunibiki* sp. nov.(Figure 4, Figure 5 and Figure 6) *Hynobius utsunomiyaorum*: Matsui et al. (2019: clade B2b3); Hayashi and Ooi (2020:100). LSID:urn:lsid:zoobank.org:pub:287B9B3A-9E5F-4F6A-BAF6-ADFC3E77B393.

Holotype. An adult male from Nishinomura, Kamionocho, Matsue City, Shimane Prefecture, Chugoku, Japan (35°29′50″ N, 132°55′11″ E; elevation: 100 m above sea level [a.s.l.]; in all cases, datum: WGS84), collected by Takayuki Iwata on 20 December 2019.

Paratype. An adult female from Nishinomura, Kamionocho, Matsue City, Shimane Prefecture, Chugoku, Japan (35° 29′ 50″ N, 132° 55′ 11″ E; elevation: 100 m a.s.l.; in all cases, datum: WGS84), collected by Takayuki Iwata on December 20, 2019. An adult male from Kamocho Chikamatsu, Unnan City, Shimane Prefecture, Chugoku, Japan (35°19′07″ N, 132°55′00″ E; elevation: 90 m a.s.l.; in all cases, datum: WGS84), collected by Takayuki Iwata on January 20, 2021.

Diagnosis. A comparatively large species (with mean SVLs of 58.9 and 58.2 mm in males and females, respectively) within Japanese lentic salamander species complex of *Hynobius*: distinct yellow stripe on dorsal and ventral edges of tail present; fifth toe of hindlimb present; DWSV absent in adult males (rarely present); distinct white spots on lateral side of body absent in adults (sometimes present); dorsal side yellowish brown to blackish brown; DGM mostly present in males; distinct black spots on dorsum absent in adults (rarely present); V-shaped vomerine teeth series; 12 or 13 (rarely 11) costal grooves; coil-shaped egg sacs.

Comparisons. The new species differs statistically from *H. utsunomiyaorum* in the following length measurements: SVL, RTAL, RMTAH, RVTL, R3FL, R5TL, and RSL in males; RAGD, RMTAH, R3TL, and R5TL in females; the lengths of these measurements, except for R3TL in females, are significantly longer in *H. kunibiki* sp. nov. than in *H. utsunomiyaorum*. The ventral edge of the tail of *H. kunibiki* sp. nov. has a distinct yellow stripe in both sexes (100%), whereas both sexes of *H. utsunomiyaorum* do not have this yellow stripe (100%). The dorsal edge of the tail of *H. kunibiki* sp. nov. has a distinct and continuous yellow stripe in both sexes (100%), whereas *H. utsunomiyaorum* males mostly do not have this yellow stripe (86.8%). *Hynobius kunibiki* sp. nov. mostly has no distinct white spots on venter in males (95.5%), but *H. utsunomiyaorum* males largely have these spots (78.9%). *Hynobius kunibiki* sp. nov. lacks distinct white spots on the lateral sides of the body in most males (95.5%) and all females (100%), whereas *H. utsunomiyaorum* males (89.5%) and females (90%) largely have these spots. *Hynobius kunibiki* sp. nov. mostly has DGM in males (81.8%), but males of *H. utsunomiyaorum* largely lack this gular mottling (76.3%). *Hynobius kunibiki* sp. nov. has the fifth toe on the hindlimbs in both sexes (100% in males and females), whereas most individuals of *H. utsunomiyaorum* do not have this fifth toe (89.5% in males, 70.0% in females). The new species is more morphologically similar to *H. setoi* than it is to *H. utsunomiyaorum* (Table 2; Figure 3), but it differs statistically from *H. setoi* in the following length measurements: RTRL, RHL, RTAL, RIND, RIOD, RSL, and RUEL in males; RVTL, RFLL, RIOD, and RSL in females; the lengths of these measurements, except for RTRL in males, were significantly longer in *H. kunibiki* sp. nov. than they were in *H. setoi*. Two lentic *Hynobius*, *H. akiensis* and *H. iwami*, are closely distributed with *H. kunibiki* sp. nov. The new species differs from *H. akiensis* by the presence of distinct yellow lines on the dorsal and ventral sides of the tail; it differs from *H. iwami* by the presence of a fifth toe [3]. Results of molecular analyses (Figure 2) [3] reveal that *H. hidamontanus* is closely related although its distribution area is largely separated from the habitat of the new species. *H. hidamontanus* differs from the new species by the absence of the fifth toe [3]. The new species and *H. sematonotos* are nearly distributed and have relatively closer relationship (Figure 2) in all Japanese *Hynobius*; however, *H. sematonotos* differs from the new species by the presence of distinct markings on the dorsum [4].

Description of holotype. A moderately large individual: HL slightly larger than HW; TAL shorter than SVL; body almost cylindrical; rounded snout; gular fold present; tail gradually compressed toward tip; expanded cloaca; webbing between digits absent; four fingers on each forelimb, order of length II = III > IV > I; five toes on each hindlimb, order of length III > IV > II > V > I; V-shaped vomerine teeth; skin smooth and shiny; scattered white spots absent on venter. The holotype had the following measurements (in mm): SVL = 60.0, TRL = 46.7, AGD = 31.2, HL = 13.6, TAL = 49.7, MTAW = 3.5, MTAH = 6.7, BTAW = 6.9, BTAH = 7.1, VTL = 3.2, VTW = 3.4, HW = 10.4, MXHW = 10.7, LFLL = 12.5, RFLL = 11.9, LHLL = 17.3, RHLL = 17.8, L1FL = 0.9, L2FL = 3.1, L3FL = 3.1, L4FL = 1.7, R1FL = 1.2, R2FL = 3.3, R3FL = 3.3, R4FL = 1.8, L1TL = 1.6, L2TL = 3.4, L3TL = 4.9, L4TL = 3.8, L5TL = 1.9, R1TL = 1.4, R2TL = 3.5, R3TL = 4.9, R4TL = 4.2, R5TL = 1.8, IND = 3.4, IOD = 3.9, LUEL = 1.5, RUEL = 1.4, SL = 4.6, LUEW = 2.9, RUEL = 2.7, LJL = 8.2, and CGN = 13.

Variation. Range, mean, and standard deviation of morphometric measurements are presented in Table 1. Morphological variations in skin markings are presented in Table 2. Males had relatively longer RHL (*t* = 2.69, *p* < 0.05), RTAL (*t* = 6.52, *p* < 0.0001), RMTAW (*t* = 3.44, *p* < 0.01), RMTAH (statistic value = −12.93, *p* < 0.0001), RHW (*t* = 2.68, *p* < 0.05), RFLL (*t* = 2.57, *p* < 0.05), RHLL (*t* = 2.12, *p* < 0.05), RSL (*t* = 2.58, *p* < 0.05), and RUEL (*t* = 3.39, *p* < 0.01) than females. However, males had relatively shorter RTRL (*t* = −2.66, *p* < 0.05) and AGD (*t* = −5.16, *p* < 0.0001) than females.

Coloration. Dorsum is uniformly blackish brown or yellowish brown without distinct black spots (rarely present); venter is lighter than dorsum with no distinct white spots in adult males (occasionally present in females); lateral side of head to tail has no distinct white spots (rarely present); iris is dark brown. When preserved, dorsal coloration tended to fade to dark gray.

Etymology. The specific name is derived from “Kunibiki″, with a mythology regarding the formation history of the Shimane Peninsula at which the type locality of the new species is located. The suggested common name in Japanese is Izumo-sanshouo.

Distribution. The new species is endemic to Shimane Prefecture and is known from Oda (including the former Oda City), Yasugi (including the former Hirose Town), Matsue (including the former Matsue City, Higashiizumo, Kashima, Shimane, Shinji, and Tamayu Towns and Yakumo Village), Izumo (including the former Hirata and Izumo Cities and Hikawa, Koryo, Sada, Taki, and Taisha Towns), and Unnan (including the former Daito, Kamo, Kisuki, and Mitoya Towns) Cities in Shimane Prefecture. The dominant vegetation type in the surrounding habitat (Figure 7) is mixed forest of chinquapin (*Castanopsis*) and live oak (*Quercus*); the new species breeds in still waters at forest edges.

Larvae and egg sacs. Larvae have distinct black dots on lateral sides of the tail and one pair of balancers during early developmental stages. Claws on the tips of fingers and toes are absent. Egg sacs are coil-shaped. From December to March, adult individuals come to ponds and attach to fallen branches or leaves in still water.

Remarks. The new species is genetically closest to *H. akiensis* and is not a monophyly with *H. utsunomiyaorum* based on previous [3] and present results.

## 5. Discussion

Based on our morphological surveys, *H. kunibiki* sp. nov. clearly did not satisfy the diagnosis suggested by Matsui et al. [3] in four ways: the fifth toe was always present, clear yellow stripes were always found on the dorsal and ventral sides of the tail, SVL was significantly larger (*p* < 0.0001), and RTAL was significantly longer (*p* < 0.0001). In particular, the presence or absence of the distinct yellow line on the ventral side of the tail was absolute and applicable to all individuals of both sexes based on our data (Table 2). Genetically, *H. utsunomiyaorum* and *H. kunibiki* sp. nov. are clearly distinguishable and do not form a monophyletic group based both on our data (Figure 2) and that of Matsui et al. [3]. Similarly, the results of phylogenetic analyses using allozyme data based on neighbor-joining and ML methods did not support the monophyly of *H. utsunomiyaorum* and *H. kunibiki* sp. nov. [6]. If *H. kunibiki* sp. nov. is a synonym of *H. utsunomiyaorum*, polyphyletic species will become valid species. According to Matsui et al. [3], *H. kunibiki* sp. nov. could not be distinguished from the clade of *H. utsunomiyaorum* based on small nucleotide polymorphism (SNP) data; hence, Matsui et al. [3] concluded that these two clades were the same species. However, this SNP-based classification is lacking *Hynobius hidamontanus* despite this species being in the same clade as *H. utsunomiyaorum*. In addition, this classification was not based on the phylogenetic and morphological species concepts; polyphyletic groups were recognized without detailed morphological comparisons. In contrast, allozyme and mitochondrial data confirm that *H. utsunomiyaorum* and *H. kunibiki* sp. nov. are two well-differentiated species [3,6]. Furthermore, *H. kunibiki* sp. nov. does not satisfy the diagnosis of *H. utsunomiyaorum* while the two species are clearly distinguishable based on our morphological data (Figure 3; Table 2). Indeed, our data suggest that this new species is more morphologically similar to *H. setoi* than to *H. utsunomiyaorum* (especially in males). Thus, *H. utsunomiyaorum* and *H. kunibiki* sp. nov. should be distinct species based on the three species concepts.

Individuals from Pop. 31 of Iinan Town (H15) collected by Matsui et al. [3] did not include the haplotype of *H. utsunomiyaorum* but did contain the haplotype of *H. kunibiki* sp. nov. (Figure 1 and Figure 2). However, this population is distantly isolated from other populations of *H. kunibiki* sp. nov. (Figure 1) and our phylogenetic analyses, including samples from Iinan Town (Pops. 106 and 107), did not support the conclusion that the new species was distributed in Iinan Town (Figure 2). Furthermore, detailed information for the voucher specimen from Iinan Town has not been provided here or elsewhere [3]. Thus, the new species we have described may not be distributed in Iinan Town; rather, the distribution range of this species may be limited to the northeastern part of Shimane Prefecture (Figure 1). *Hynobius kunibiki* sp. nov. is parapatrically distributed with *H. setoi* and *H. iwami* (Figure 1), but analyses of nuclear and mitochondrial DNA suggest that *H. kunibiki* sp. nov. is clearly separated from these two species genetically (Figure 2) [3]. Additionally, *H. iwami* has no fifth toe; thus, it is clearly separated from *H. kunibiki* sp. nov. both genetically and morphologically [3].

According to Matsui et al. [3], *H. setoi* is found at the lowland areas along the Japan Sea from northwestern Hyogo Prefecture to the northeastern part of Shimane Prefecture. Based on our analyses, haplotypes of *H. setoi* were found from the westernmost part of Hyogo, Tottori, from west to east, and the easternmost part of Shimane Prefectures; however, these haplotypes were not detected from around the northeastern part of Shimane Prefecture (Figure 1 and Figure 2). Similar to our findings, Hayashi and Ooi [5] could not find the haplotypes of *H. setoi* in the northeastern part of Shimane Prefecture, so individuals of *H. setoi* from Daitocho Sannoji, Shinobuchi, and Hirosecho Shinoyamasa surveyed by Matsui et al. [3] are unlikely to be *H. setoi*; thus, the analyses of Matsui et al. [3] may have combined two different species. Consequently, the diagnosis of *H. setoi* should be carefully applied when attempting to identify this species. Actually, the mean SVL, FLL/SVL, and HLL/SVL suggested by Matsui et al. [3] did not differ significantly between *H. setoi* and *H. kunibiki* sp. nov., and both species possessed distinct yellow stripes on the dorsal and ventral sides of their tails (Table 2); therefore, these two species cannot be separated according to the morphology described in the diagnosis of *H. setoi* by Matsui et al. [3]. In fact, male *H. kunibiki* sp. nov. were closer to *H. setoi* than to *H. utsunomiyaorum* in terms of Euclidean distance, despite *H. kunibiki* sp. nov. having a closer genetic relationship with *H. utsunomiyaorum* than it has with *H. setoi* (Figure 2). Males of *H. kunibiki* sp. nov. and *H. setoi* differed significantly in terms of RTRL, RHL, RTAL, RIND, RIOD, RSL, and RUEL; hence, these morphological characteristics will be a useful reference by which to distinguish the two species.

Consistent with the results of Matsui et al. [3], our phylogenetic analyses suggested that *H. hidamontanus* is a synonym of *H. utsunomiyaorum* (Figure 2). However, we could not collect samples of *H. hidamontanus* because it is strictly protected by law in Japan. Therefore, a morphological survey of this species will be required for its synonymization. In the present study, the entire distribution ranges of *H. kunibiki* sp. nov. and *H. setoi* were revealed in detail (Appendix A; Figure 1), whereas the distribution ranges of *H. utsunomiyaorum* from the central and eastern part of Hiroshima Prefecture and *H. iwami* from the western part of Shimane Prefecture were not revealed in detail. Thus, additional research will be required to determine the distribution areas of the latter two species.

We conducted phylogenetic analyses using all Japanese *Hynobius* species (Figure 2). According to Dubois and Raffaëlli [19], Japanese *Hynobius* are divided into five supraspecies (i.e., *hidamontanus*, *lichenatus*, *naevius*, *nebulosus*, and *stejnegeri*); however, supraspecies *hidamontanus* and *naevius* cannot be distinguished genetically (Figure 2). If subgenera of *Hynobius* were to be divided by supraspecies, four supraspecies would be recognized based on BI: the *Hynobius nebulosus* supraspecies (including *H. bakan*, *H. dunni*, *H. iwami*, *H. nebulosus*, *H. okiensis*, and *H. tsuensis*), *Hynobius lichenatus* supraspecies (containing *H. abei*, *H. lichenatus*, *H. mikawaensis*, *H. nigrescens*, *H. setoi*, *H. setouchi*, *H. takedai*, *H. tokyoensis*, and *H. vandenburghi*), *Hynobius shinichisatoi* supraspecies (including *H. amakusaensis*, *H. ikioi*, *H. osumiensis*, and *H. shinichisatoi*), and *Hynobius naevius* supraspecies (comprising *H. abuensis*, *H. akiensis*, *H. guttatus*, *H. hidamontanus*, *H. hirosei*, *H. katoi*, *H. kuishiensis*, *H. kunibiki* sp. nov., *H. naevius*, *H. oyamai*, *H. sematonotos*, *H. stejnegeri*, *H. tosashimizuensis*, *H. tsurugiensis*, and *H. utsunomiyaorum*). Dubois and Raffaëlli [19] used the term supraspecies, but the common morphological characteristics of each supraspecies have not yet been evaluated to the best of our knowledge; hence, this term should not be used without an initial morphological definition. If the supraspecies are valid, further morphological studies among the groups are necessary.

Following this description, the number of Japanese *Hynobius* spp. is 38, but some taxonomic problems have been left unsolved. Matsui et al. [3] suggested that *H. akiensis* and *H. nebulosus* are not monophyletic species, and they may be divided into two species. Further, the study only described male morphology; thus, female morphology of many *Hynobius* species remains unidentified. The task of clarifying the distribution ranges of lentic *Hynobius* also needs to be undertaken as detailed distribution ranges of *H. setouchi*, *H. akiensis*, and *H. vandenburghi* are still unknown. Future studies involving taxonomic reexamination and field surveys to clarify the distribution ranges of both sexes of the Japanese *Hynobius* species are required.

The localities of the four species used in this study have been provided in detail, which will support conservation activities. In particular, *H. kunibiki* sp. nov. and *H. setoi* are morphologically similar, so definitive identification of these species will be difficult where DNA data is unavailable. The information in this study is expected to be useful for confirming the distribution ranges of these two species. However, it is possible that overcollection could decimate the *Hynobius* species following the release of detailed locality information. Indeed, overfishing of *H. utsunomiyaorum* has already been recognized [20]. Therefore, the conservation status of *H. kunibiki* sp. nov. must be reassessed, and management plans for its conservation and regulation by national or regional administrations are immediately required to prevent overcollection and possible extinction.

## 6. Conclusions

The monophyly of the Izumo Lineage of *Hynobius utsunomiyaorum* and *H. utsunomiyaorum* was rejected by the molecular evidence. Besides the genetic evidence, the Izumo Lineage of *H. utsunomiyaorum* did not satisfy the diagnosis from the original description of *H. utsunomiyaorum* in morphology. Furthermore, significant morphological differences between them were detected in both sexes, so we described the Izumo Lineage of *H. utsunomiyaorum* as *H. kunibiki* based on morphological, phylogenetic, and evolutionary species concepts. This new species is limited to the northern part of Shimane Prefecture, and the conservation status should be reassessed for appropriate conservation activities.

## Figures and Tables

**Figure 1 animals-11-02187-f001:**
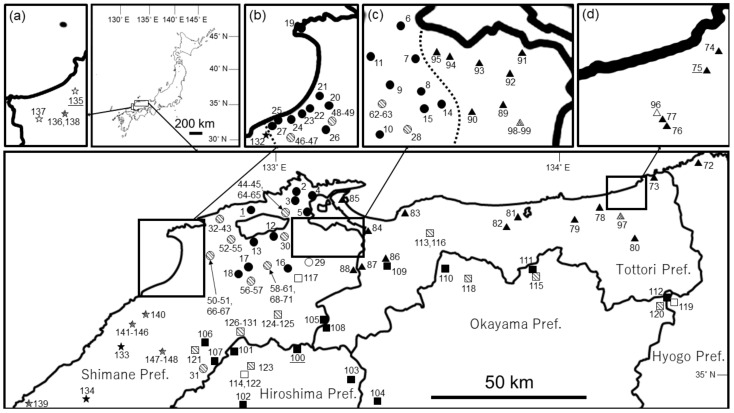
Localities for populations sampled in northern Chugoku, Japan. Population numbers match those used for molecular analyses (see Appendix A and Figure 2). Lower mapped area includes the northern part of Chugoku (Tottori, central to eastern part of Shimane, northern part of Okayama, and northeastern part of Hiroshima Prefectures) and northwestern part of Kinki (Hyogo Prefecture). Top left magnified area (**a**) includes the westernmost part of Shimane Prefecture; (**b**) includes the north central part of Shimane Prefecture; (**c**) includes the easternmost part of Shimane Prefecture; (**d**) includes the eastern part of Tottori Prefecture. The closed symbols correspond to each of four species sampled in this study: *H. kunibiki* sp. nov. (closed circles), *H. setoi* (closed triangles), *H. utsunomiyaorum* (closed squares), and *H. iwami* (closed stars). The open symbols correspond to each of four species cited from Matsui et al. [3]: *H. kunibiki* sp. nov. (open circle), *H. setoi* (open triangle), *H. utsunomiyaorum* (open squares), and *H. iwami* (open stars). The open symbols including diagonal lines correspond to each of four species cited from Matsui et al. [3] or Hayashi and Ooi [5] for which the detailed locality is unclear: *H. kunibiki* sp. nov. (shadow circles), *H. setoi* (shadow triangles), *H. utsunomiyaorum* (shadow squares), and *H. iwami* (shadow stars). For the morphological comparisons, individuals of *H. kunibiki* sp. nov., *H. setoi*, and *H. utsunomiyaorum* were sampled from several localities including all major fragmented distribution areas: *H. kunibiki* sp. nov. from Pops. 1 (male = 14, female = 6), 2 (male = 2, female = 0), 3 (male = 1, female = 1), 8 (male = 2, female = 0), 17 (male = 2, female = 1), 22 (male = 1, female = 0), and 23 (male = 0, female = 1); *H. setoi* from Pops. 75 (male = 9, female = 4), 76 (male = 8, female = 2), 81 (male = 5, female = 1), 82 (male = 2, female = 2), 84 (male = 7, female = 5), and 87 (male = 0, female = 1); and *H. utsunomiyaorum* from Pops. 100 (male = 24, female = 7), 102 (male = 5, female = 1), 111 (male = 4, female = 1), and 112 (male = 5, female = 1). Underlined labels (Pops. 1, 75, 100, and 135) show the type locality of each species.

**Figure 2 animals-11-02187-f002:**
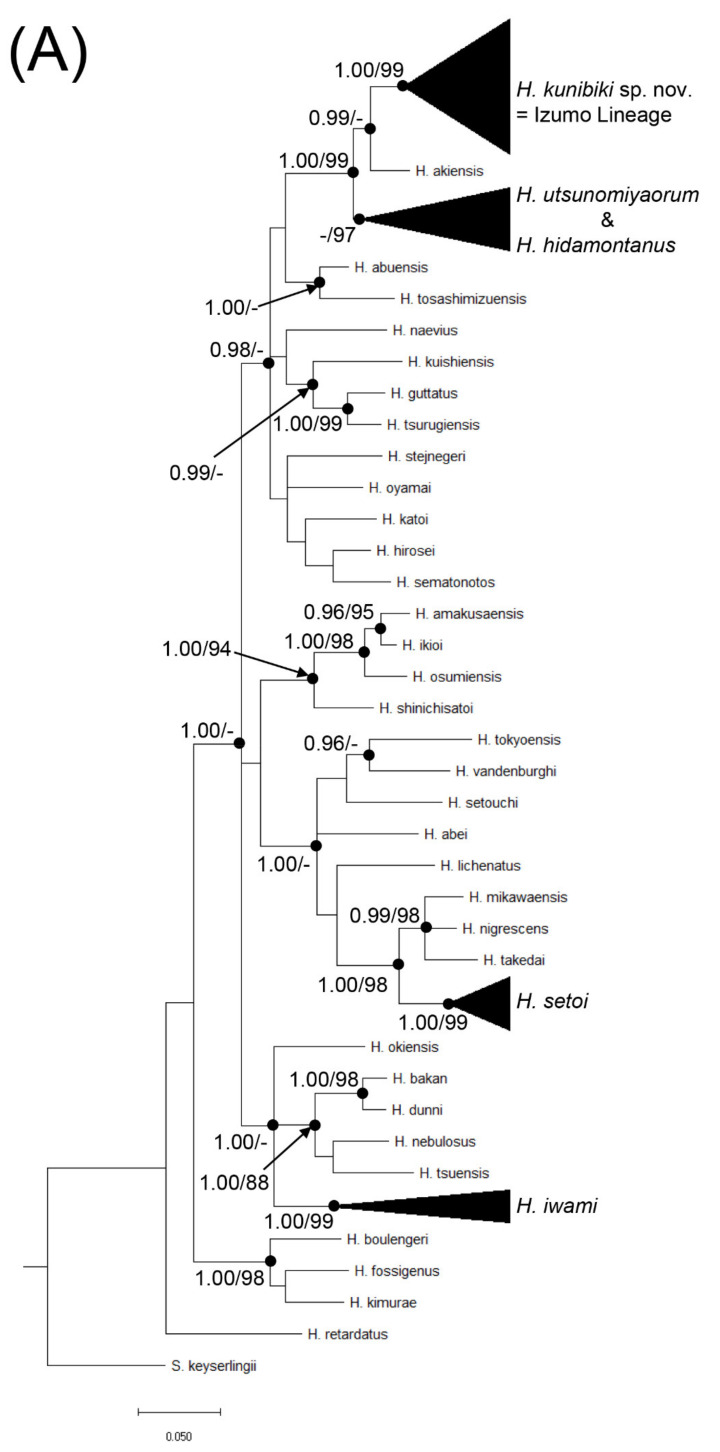
Bayesian inference phylogenetic tree based on 630-base pair (bp) cytochrome *b* sequences rooted with *Salamandrella keyserlingii*. Five phylogenies indicate that phylogenetic relationships of (**A**) all Japanese *Hynobius* species, (**B**) within *H. kunibiki* sp. nov., (**C**) within *H. utunomiyaorum* (also *H. hidamontanus*), (**D**) within *H. setoi*, and (**E**) within *H. iwami*, respectively. Scale bars indicate the genetic distance (expected changes per site). Numbers near nodes indicate the supported level based on Bayesian posterior probabilities (BPP) and bootstrap values (BSV), respectively. Underlined labels indicate the type locality of the three species and *H. iwami*. Numbers appearing in parentheses after the labels correspond to population localities as indicated in Appendix A and Figure 1.

**Figure 3 animals-11-02187-f003:**
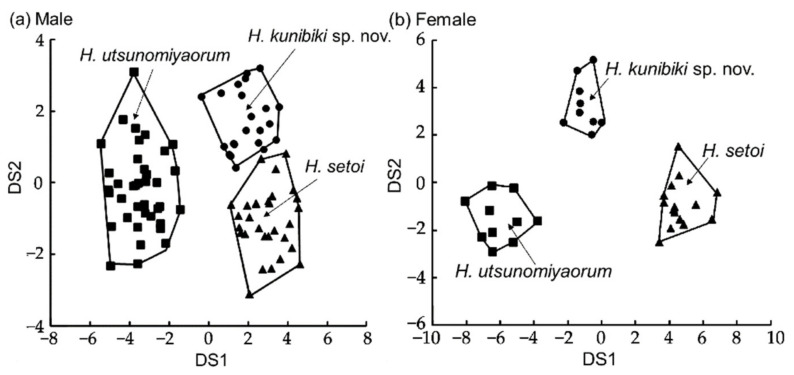
Two-dimensional plots based on canonical discriminant analysis. The *x* and *y* axes indicate discriminant score 1 (DS1) and discriminant score 2 (DS2), respectively. The contribution ratios of DS1 and DS2 in males and females were as follows: DS1: 88.90% for males, 83.96% for females; DS2: 11.10% for males, 16.04% for females. Circles, triangles, and squares indicate scores for individuals from the type localities of *Hynobius kunibiki* sp. nov., *H. setoi*, and *H. utsunomiyaorum*, respectively.

**Figure 4 animals-11-02187-f004:**
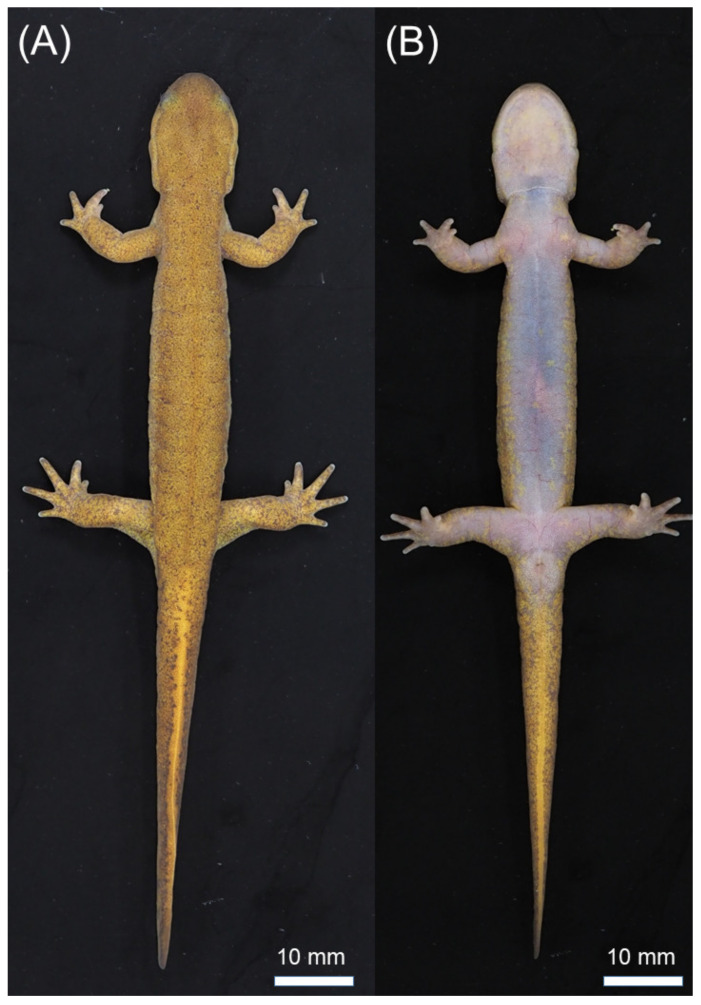
Holotype of *Hynobius kunibiki* sp. nov. (TRPM-ARA-0000014, adult male): (**A**) dorsal and (**B**) ventral views.

**Figure 5 animals-11-02187-f005:**
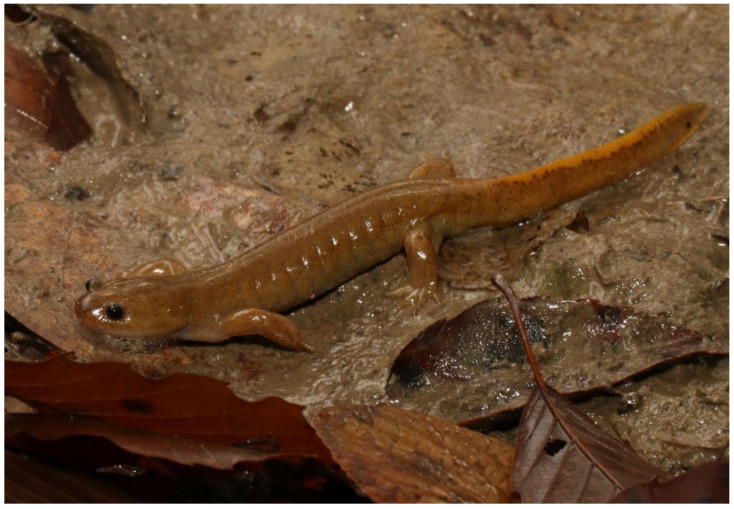
Live holotype of *Hynobius kunibiki* sp. nov. (TRPM-ARA-0000014, adult male).

**Figure 6 animals-11-02187-f006:**
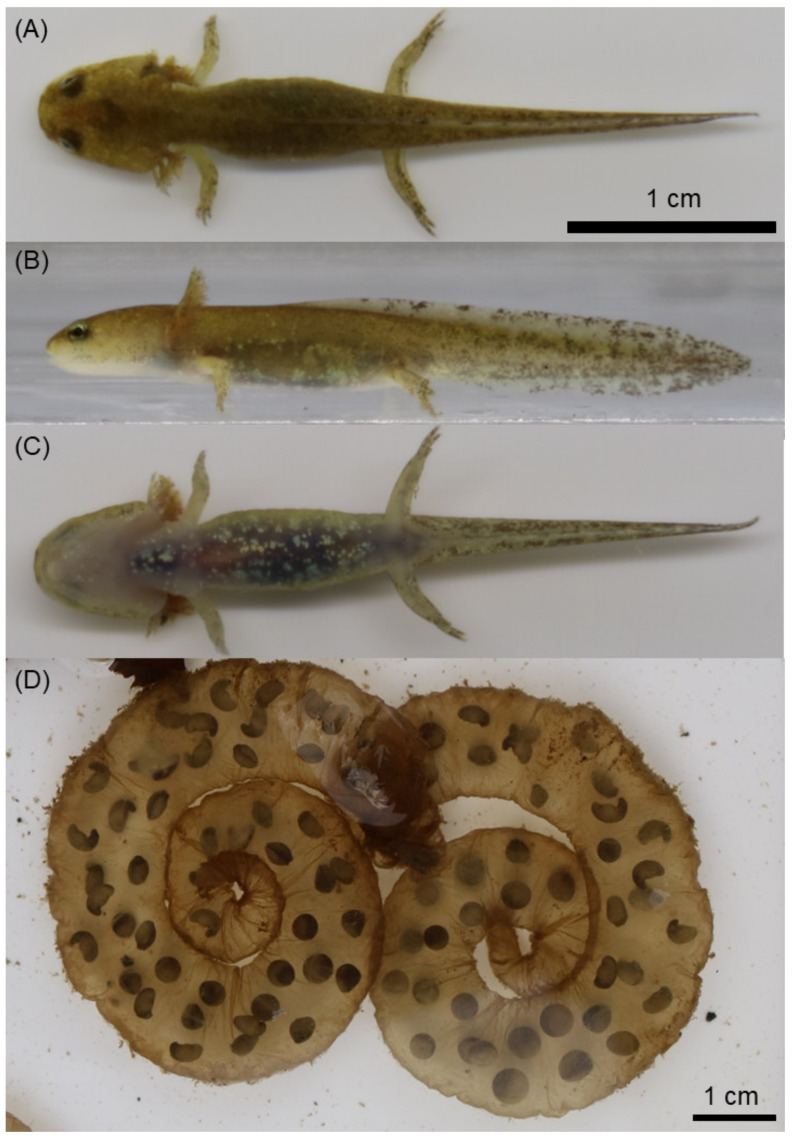
Larva of *Hynobius kunibiki* sp. nov.: (**A**) dorsal, (**B**) lateral, and (**C**) ventral sides, as well as (**D**) egg sacs.

**Figure 7 animals-11-02187-f007:**
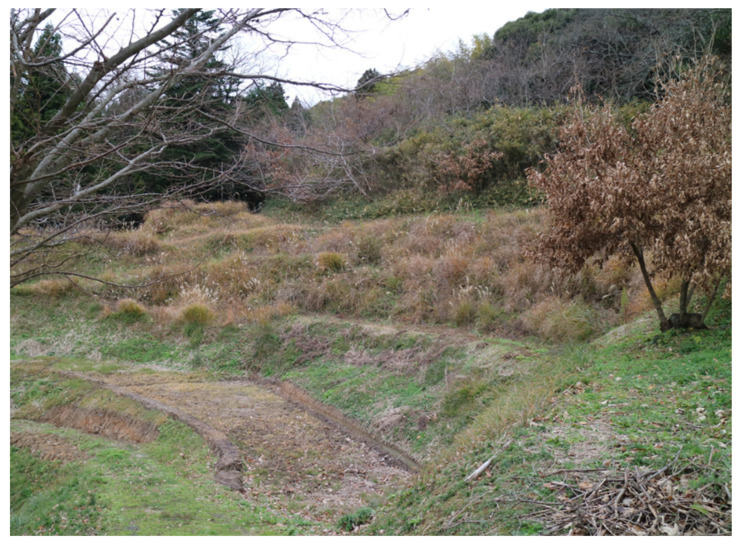
Habitat at the type locality of *Hynobius kunibiki* sp. nov. in Nishinomura, Kamionocho, Matsue City, Shimane Prefecture.

**Table 1 animals-11-02187-t001:** Measurements (mm) of snout–vent length (SVL) and character ratios (R = %SVL) of trunk length (TRL) to lower jaw length (LJL) of the three species of *Hynobius* collected in 2018–2021 (*n* = number of individuals). Values are reported as means ± standard deviation. Ranges are shown in parentheses. See Materials and Methods section for definitions of morphological traits.

	*H. kunibiki* sp. nov.		*H. setoi*		*H. utsunomiyaorum*	
	Male	Female	Male	Female	Male	Female
Trait	*n* = 22	*n* = 9	*n* = 30	*n* = 15	*n* = 38	*n* = 10
SVL	58.9 ± 3.15	58.2 ± 4.24	60.7 ± 6.37	63.0 ± 6.16	53.4 ± 4.10	54.6 ± 4.19
	(52.7–63.9)	(51.0–66.6)	(49.0–70.8)	(50.5–72.5)	(40.0–59.8)	(48.0–60.5)
RTRL	76.5 ± 1.00	77.6 ± 1.24	77.4 ± 1.00	77.7 ± 2.20	76.7 ± 1.65	77.0 ± 0.62
	(75.7–78.4)	(75.6–79.2)	(75.7–80.1)	(71.0–80.8)	(74.3–84.8)	(76.2–77.8)
RAGD	51.3 ± 2.06	55.7 ± 2.35	52.3 ± 1.72	55.5 ± 2.63	52.1 ± 1.68	52.8 ± 2.27
	(47.7–54.9)	(50.8–59.2)	(48.1–55.9)	(50.7–59.1)	(48.9–56.5)	(49.0–55.9)
RHL	24.2 ± 0.84	23.3 ± 0.91	23.5 ± 0.99	23.0 ± 1.10	24.4 ± 1.03	24.2 ± 0.73
	(22.6–25.7)	(21.8–24.6)	(22.0–26.1)	(21.2–24.8)	(22.0–26.5)	(22.6–25.0)
RTAL	82.0 ± 4.09	71.3 ± 4.28	78.3 ± 6.15	69.5 ± 5.47	73.1 ± 7.28	70.5 ± 6.51
	(74.1–93.1)	(62.8–76.4)	(67.5–94.7)	(59.7–77.0)	(52.9–86.0)	(58.8–81.4)
RMTAW	6.9 ± 0.84	5.8 ± 0.76	6.6 ± 1.01	6.5 ± 0.87	6.7 ± 0.96	6.6 ± 0.59
	(5.5–8.3)	(4.9–7.4)	(4.8–8.8)	(4.8–7.7)	(4.4–8.8)	(5.7–7.5)
RMTAH	13.0 ± 0.95	10.8 ± 1.38	13.0 ± 1.36	11.3 ± 1.17	10.2 ± 1.37	9.3 ± 0.75
	(11.4–14.6)	(7.5–12.2)	(10.8–15.9)	(9.1–13.7)	(6.8–12.9)	(8.3–10.6)
RVTL	5.1 ± 0.46	5.3 ± 0.50	4.9 ± 0.48	4.6 ± 0.63	4.5 ± 0.50	4.7 ± 0.38
	(4.3–6.1)	(4.3–5.9)	(3.9–6.1)	(3.5–5.6)	(3.6–5.9)	(4.1–5.4)
RVTW	5.4 ± 0.51	5.4 ± 0.48	5.4 ± 0.47	5.1 ± 0.61	5.7 ± 0.50	5.6 ± 0.28
	(4.4–6.3)	(4.7–6.0)	(4.4–6.1)	(4.3–6.4)	(4.7–6.7)	(5.0–6.0)
RHW	17.7 ± 0.62	17.0 ± 0.87	17.5 ± 0.87	16.6 ± 0.77	17.3 ± 0.73	16.7 ± 0.71
	(16.6–19.2)	(15.8–18.2)	(16.2–19.0)	(15.3–17.6)	(15.7–18.8)	(15.5–18.2)
RFLL	25.5 ± 1.49	24.0 ± 1.45	24.8 ± 1.71	22.3 ± 0.92	24.8 ± 1.69	25.0 ± 2.50
	(23.1–29.5)	(21.7–26.1)	(22.6–30.6)	(20.7–24.2)	(21.0–27.9)	(21.1–28.8)
RHLL	32.4 ± 1.60	31.0 ± 1.87	32.1 ± 1.31	29.4 ± 1.75	31.8 ± 1.30	31.8 ± 1.50
	(29.9–35.2)	(27.7–34.8)	(29.5–36.3)	(25.9–31.9)	(29.3–35.6)	(29.3–34.2)
R2FL	5.2 ± 0.62	4.7 ± 0.69	4.9 ± 0.71	4.3 ± 0.68	5.1 ± 0.71	4.9 ± 0.65
	(3.9–6.3)	(4.1–6.0)	(2.9–6.5)	(3.1–5.5)	(2.9–6.6)	(4.0–5.9)
R3FL	4.1 ± 0.54	3.7 ± 0.87	4.4 ± 0.67	4.0 ± 0.58	3.6 ± 0.67	3.6 ± 0.58
	(2.5–4.9)	(2.3–5.0)	(2.8–5.5)	(3.0–5.1)	(1.4–4.9)	(2.8–4.7)
R3TL	7.9 ± 0.64	7.5 ± 0.61	8.0 ± 0.56	7.3 ± 0.84	8.3 ± 0.82	8.4 ± 0.59
	(7.0–9.1)	(6.4–8.3)	(7.0–9.1)	(5.9–9.1)	(5.5–10.4)	(7.4–9.4)
R5TL	1.9 ± 0.62	1.8 ± 0.69	2.2 ± 0.62	2.3 ± 0.69	0.2 ± 0.48	0.3 ± 0.48
	(0.3–3.0)	(0.5–2.6)	(1.1–3.3)	(1.1–3.5)	(0.0–2.2)	(0.0–1.2)
RIND	5.3 ± 0.57	4.9 ± 0.52	4.7 ± 0.48	4.6 ± 0.29	5.2 ± 0.45	4.9 ± 0.67
	(3.9–6.3)	(4.2–6.1)	(3.7–5.7)	(4.1–5.2)	(4.4–6.1)	(4.3–6.3)
RIOD	6.4 ± 0.51	6.0 ± 0.37	5.9 ± 0.63	5.6 ± 0.38	6.3 ± 0.38	6.1 ± 0.46
	(5.3–7.5)	(5.6–6.6)	(4.9–7.1)	(5.0–6.3)	(5.4–6.9)	(5.5–7.0)
RUEL	3.2 ± 0.25	3.2 ± 0.19	3.2 ± 0.30	3.2 ± 0.28	3.3 ± 0.28	3.3 ± 0.27
	(2.8–3.8)	(2.9–3.4)	(2.5–3.9)	(2.8–3.8)	(2.8–3.9)	(2.9–3.8)
RSL	6.7 ± 0.32	6.4 ± 0.35	6.0 ± 0.34	5.7 ± 0.29	6.4 ± 0.38	6.1 ± 0.53
	(6.1–7.3)	(5.7–6.8)	(5.2–6.7)	(5.3–6.3)	(5.4–7.4)	(5.0–6.6)
RUEW	4.6 ± 0.20	4.3 ± 0.29	4.4 ± 0.28	4.2 ± 0.29	4.6 ± 0.37	4.4 ± 0.25
	(4.2–5.1)	(3.9–4.7)	(3.8–4.9)	(3.7–5.0)	(4.1–5.5)	(4.1–4.9)
RLJL	13.9 ± 0.69	13.7 ± 0.69	13.4 ± 0.75	13.1 ± 0.85	14.1 ± 0.73	13.4 ± 0.61
	(12.9–15.7)	(13.1–15.4)	(11.5–14.9)	(11.5–14.6)	(12.7–16.4)	(12.2–14.1)

**Table 2 animals-11-02187-t002:** Characteristics of skin markings among the three species of *Hynobius*. The values indicate the number of individuals exhibiting that characteristic with percentages related to each condition in parentheses. See Materials and Methods section for definitions of morphological characteristics.

		*H. kunibiki* sp. nov.		*H. setoi*		*H. utsunomiyaorum*	
		Male	Female	Male	Female	Male	Female
Character	Condition	*n* = 22	*n* = 9	*n* = 30	*n* = 15	*n* = 38	*n* = 10
DWSV	Absent	21 (95.5%)	5 (55.6%)	21 (70.0%)	7 (46.7%)	8 (21.1%)	1 (10.0%)
	Present	1 (4.5%)	4 (44.4%)	9 (30.0%)	8 (53.3%)	30 (78.9%)	9 (90.0%)
DWSL	Absent	21 (95.5%)	9 (100%)	28 (93.3%)	13 (86.7%)	4 (10.5%)	1 (10.0%)
	Present	1 (4.5%)	0 (0%)	2 (6.7%)	2 (13.3%)	34 (89.5%)	9 (90.0%)
DCTYLD	Absent	0 (0%)	0 (0%)	0 (0%)	0 (0%)	33 (86.8%)	6 (60.0%)
	Present	22 (100%)	9 (100%)	30 (100%)	15 (100%)	5 (13.2%)	4 (40.0%)
DTYLV	Absent	0 (0%)	0 (0%)	0 (0%)	0 (0%)	38 (100%)	10 (100%)
	Present	22 (100%)	9 (100%)	30 (100%)	15 (100%)	0 (0%)	0 (0%)
DGM	Absent	4 (18.2%)	9 (100%)	14 (46.7%)	15 (100%)	29 (76.3%)	10 (100%)
	Present	18 (81.8%)	0 (0%)	16 (53.3%)	0 (0%)	9 (23.7%)	0 (0%)
CGN	11	1 (4.5%)	1 (11.1%)	2 (6.7%)	0 (0%)	7 (18.4%)	1 (10.0%)
	12	18 (81.8%)	4 (44.4%)	24 (80.0%)	7 (46.7%)	23 (60.5%)	8 (80.0%)
	13	3 (13.6%)	4 (44.4%)	4 (13.3%)	8 (53.3%)	8 (21.1%)	1 (10.0%)
CFBALN	2.0	0 (0%)	0 (0%)	1 (3.3%)	0 (0%)	0 (0%)	0 (0%)
	1.5	1 (4.5%)	0 (0%)	0 (0%)	0 (0%)	0 (0%)	0 (0%)
	1.0	5 (22.7%)	0 (0%)	2 (6.7%)	0 (0%)	3 (7.9%)	0 (0%)
	0.5	4 (18.2%)	2 (22.2%)	2 (6.7%)	0 (0%)	7 (18.4%)	1 (10.0%)
	0.0	6 (27.3%)	0 (0%)	8 (26.7%)	0 (0%)	9 (23.7%)	3 (30.0%)
	−0.5	4 (18.2%)	0 (0%)	5 (16.7%)	0 (0%)	9 (23.7%)	1 (10.0%)
	−1.0	2 (9.1%)	3 (33.3%)	5 (16.7%)	0 (0%)	8 (21.1%)	3 (30.0%)
	−1.5	0 (0%)	0 (0%)	5 (16.7%)	2 (13.3%)	1 (2.6%)	1 (10.0%)
	−2.0	0 (0%)	3 (33.3%)	2 (6.7%)	8 (53.3%)	1 (2.6%)	1 (10.0%)
	−2.5	0 (0%)	0 (0%)	0 (0%)	2 (13.3%)	0 (0%)	0 (0%)
	−3.0	0 (0%)	1 (11.1%)	0 (0%)	3 (20.0%)	0 (0%)	0 (0%)

## Data Availability

The data presented in this paper are available on request from the corresponding author.

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
