# Peer review of "Taxonomic Reassessment of the Izumo Lineage of Hynobius utsunomiyaorum: Description of a New Species from Chugoku, Japan"

_animals, 2021, doi:10.3390/ani11082187_

Round 1
Reviewer 1 Report
This article is very interesting as it brings another piece to the taxonomic puzzle of Hynobius salamanders. Congratulations to the authors for managing to collect such a large sample, more than 120 salamanders have been measured for mophological analysis. This is encouraging as integrative taxonomy is mostly required nowadays in taxonomic revisions and description of new species. Thanks to this huge effort, the authors describe a new species of Hynobius that is supported by genetic and morphometric data.
I have some remarks (minor corrections) though as I think the species description could benefit from some additional informations. I found that you focus on four species based on their range, but actually information on more species would have been expected. Here are some comments on specific topics that would need to be amended in my opinion, as well as some minor corrections I found throughout the text.
From what I understand, there are at least seven species of Hynobius in the Chugoku district: H. bakan, H. abuensis, H. iwami, H. utsunomiyaorum, H. setoi, H. akiensis and H. setouchi. I understand that you focus your study on the northern part of the district, but then, why did you omit H. iwami from the morphological study? I would have expected to find some comparisons with this species as it occurs at close range from H. utsunomiyaorum at least. If you did not gather yourself samples from this species, maybe it is possible to collect data from literature. It seems to me that such data are available in Matsui et al. 2019, so it should be possible to add them to complete the data set.
Actually, for the morphometrics analysis, it would even have been better to only include specimens belonging to all species of the utsunomiyaorum clade (therefore discard H. setoi), in order to test the congruence between morphometrics and genetics. I would therefore recommend to perform the canonical analysis only with H. kubini, H. akiensis, H. utsunomiyaorum, and H. hidamontanus. You can also include H. setoi in this analysis, considering the morphological similarity between H. kubini and H. setoi as mentionned lines 81-85, but in my opinion this is not really necessary as both species are clearly well distingished based on genetic data and that the morphological characters that you use clearly distinguish them also, as shown in the diagnosis of H. kubini.
Comparisons paragraph in the description: Normally, it would be expected to compare the features of the new species with all other species in the genus in order to get a precise and exhaustive view of its singularity. Here, you only compare it with three other species in the comparison section. This is not satisfactory and should be amended accordingly. Comparisons with other species require a close examination of specimens kept in museum, or data previously published. Such previously published data is available for several species (Matsui et al. 2019 for example), and further measurements from museum specimens should enable to complete the data set. It is possible to group some comparisons, for example if a group of Hynobius from different localities share a striking feature that is very different from the ones you study (for example different range, different size, different shapes, longer limbs, etc.).
line 48: replace “species in” by “to”
line 52: replace “; however, several cryptic species are probably” by “that are”
line 70-71: replace “clade does not form a monophyly with” by “lineage paraphyletic with respect to”
line 137: replace “monophyly” by “monophyletic”
Maybe Table 1 should be provided as supplementary material?
Figure 1: write the name of the species in italics
Line 236: replace “formed a monophyletic group” by "do”
Libe 237: replace “with the clade including” by “, which includes”
Line 249: the legend of this figure is missing
Figure 2: This figure should be split as it is not clear to which figure the legend refers to. The figures should be labeled with (A), (B), (C), and so on, and the legend should clearly explain what each figure represent. Or the figures should be presented as different figures (Fig. 2, Fig. 3 and so on). For me both solutions would work, but this should be modified in either way.
Line 459: replace “not monophyletic” by “two well-differentiated”
Lines 569-576: the conclusion is redundant with the introduction and adds nothing to the whole text. In my opinion, this part should be removed.
I could not get hold of the following reference: Hayashi, M.; Ooi, K. Molecular Study on Hynobius in East Area of Shimane Prefecture, Japan: Notes on Haplotype Diversity of mtDNA and Their Distribution. Bull. Hoshizaki Green Found. 2020, 23, 99–104. Where is it possible to find it ? It is not referenced in google nor in google scholar or researchgate. Is it an article in Japanese? If so, maybe it should be relevant to mention it in the reference list.
Author Response
Dear Reviewer 1,
Thank you for your critical comments.
We respond to your comments with the following.
Point 1
From what I understand, there are at least seven species of Hynobius in the Chugoku district: H. bakan, H. abuensis, H. iwami, H. utsunomiyaorum, H. setoi, H. akiensis and H. setouchi. I understand that you focus your study on the northern part of the district, but then, why did you omit H. iwami from the morphological study? I would have expected to find some comparisons with this species as it occurs at close range from H. utsunomiyaorum at least. If you did not gather yourself samples from this species, maybe it is possible to collect data from literature. It seems to me that such data are available in Matsui et al. 2019, so it should be possible to add them to complete the data set.
Response 1
We did not include Hynobius iwami because this species is clearly distinguished from H. kunibiki, H. utsunomiyaorum, and H. setoi in morphology, so statistical analyses for identification of this species are unnecessary. In addition, H. iwami was clearly distinguished not only morphology, but also mitochondrial and nuclear DNA by Matsui et al. (2019). Thus, we removed this species in morphological analyses. However, we agree with your opinion because the best comparison for description is conducted using all Hynobius species as possible. Now, we have no enough samples of H. iwami for statistical analyses, and we cannot compare the two species based on statistical analyses (H. kunibiki and H. iwami), unfortunately. Instead, we compared the two species (H. kunibiki and H. iwami) based on the literature data from Matsui et al. (2019). We added some sentences into the subsection “Comparisons” as follows (after Line 374): Two lentic Hynobius, H. akiensis and H. iwami, are closely distributed with H. kunibiki sp. nov.. The new species clearly differs from H. akiensis by the presence of distinct yellow lines on dorsal and ventral sides of the tail; it differs from H. iwami by the presence of a fifth toe (Matsui et al. 2019). Results of molecular analyses (Fig. 2) (Matsui et al. 2019) revealed that H. hidamontanus is closely related although its distribution area is largely separated from the habitat of the new species. H. hidamontanus differs from the new species by the absence of the fifth toe. The new species and H. sematonotos are nearly distributed and have relatively closer relationship (Fig. 2) in all Japanese Hynobius; however, H. sematonotos differs from the new species by the presence of distinct markings on the dorsum (Tominaga et al. 2019).
Point 2
Actually, for the morphometrics analysis, it would even have been better to only include specimens belonging to all species of the utsunomiyaorum clade (therefore discard H. setoi), in order to test the congruence between morphometrics and genetics. I would therefore recommend to perform the canonical analysis only with H. kubini, H. akiensis, H. utsunomiyaorum, and H. hidamontanus. You can also include H. setoi in this analysis, considering the morphological similarity between H. kunibiki and H. setoi as mentionned lines 81-85, but in my opinion this is not really necessary as both species are clearly well distingished based on genetic data and that the morphological characters that you use clearly distinguish them also, as shown in the diagnosis of H. kunibiki.
Response 2
We agree with your opinion because the morphological comparison with more two species (H. akiensis and H. hidamontanus) is the best analysis. However, these two species are strictly protected by the law of local governments. We tried to get the collecting permission to local government, but we cannot get the collecting permission of H. akiensis and H. hidamontanus. In the future morphological studies, we will continuously try to get the collecting permission of these species to local governments. Instead, we compared between H. kunibiki and H. akiensis, H. kunibiki and H. iwami, and H. kunibiki and H. hidamontanus based on the morphological data from Matsui et al. 2019. We added the several sentences in the subsection “Comparisons”, and please refer to the response of Point 1.
Point 3
Comparisons paragraph in the description: Normally, it would be expected to compare the features of the new species with all other species in the genus in order to get a precise and exhaustive view of its singularity. Here, you only compare it with three other species in the comparison section. This is not satisfactory and should be amended accordingly. Comparisons with other species require a close examination of specimens kept in museum, or data previously published. Such previously published data is available for several species (Matsui et al. 2019 for example), and further measurements from museum specimens should enable to complete the data set. It is possible to group some comparisons, for example if a group of Hynobius from different localities share a striking feature that is very different from the ones you study (for example different range, different size, different shapes, longer limbs, etc.).
Response 3
We agree with your opinion, but we did not perform the survey of specimens in museums. Instead, we compared H. kunibiki and other species that distributed in the near range with the new species or have a closer relationship in genetics (H. kunibiki vs. H. akiensis, H. kunibiki vs. H. iwami, H. kunibiki vs. H. hidamontanus, and H. kunibiki vs. H. sematonotos) based on the morphological data from Matsui et al. (2019) or Tominaga et al. (2019). We added the several sentences in the subsection “Comparisons”, and please refer to the response of Point 1.
Point 4
line 48: replace “species in” by “to”
Response 4
We replaced it as follows: 37 (67.3%) of which are endemic to Japan (Frost 2021).
Point 5
line 52: replace “; however, several cryptic species are probably” by “that are”
Response 5
We replaced it as follows: that are yet to be described (Matsui et al. 2019; Hayashi and Ooi 2020).
Point 6
line 70-71: replace “clade does not form a monophyly with” by “lineage paraphyletic with respect to”
Response 6
We replaced it as follows: this lineage paraphyletic with respect to H. utsunomiyaorum (Matsui et al. 2006; Matsui et al. 2019)
Point 7
line 137: replace “monophyly” by “monophyletic”
Response 7
We replaced it as follows: We regarded a group as monophyletic when a node had a posterior probability ≥0.95 according to the criterion of Huelsenbeck and Rannala (2004).
Point 8
Maybe Table 1 should be provided as supplementary material?
Response 8
We changed and provided the table as supplementary material.
Point 9
Figure 1: write the name of the species in italics
Response 9
We rewrote them in italics.
Point 10
Line 236: replace “formed a monophyletic group” by "do”
Response 10
We replaced it as follows: whereas the former group and Hynobius akiensis do (Fig. 2)
Point 11
Libe 237: replace “with the clade including” by “, which includes”
Response 11
We replaced it as follows: this species formed a monophyletic group which includes Hynobius mikawaensis, Hynobius takedai, and Hynobius nigrescens (Fig. 2).
.
Point 12
Line 249: the legend of this figure is missing
Figure 2: This figure should be split as it is not clear to which figure the legend refers to. The figures should be labeled with (A), (B), (C), and so on, and the legend should clearly explain what each figure represent. Or the figures should be presented as different figures (Fig. 2, Fig. 3 and so on). For me both solutions would work, but this should be modified in either way.
Response 12
We labeled these figures, and explanations of each figure were added as follows (we also revised Line253–257): Five phylogenies indicate that phylogenetic relationships of (A) all Japanese Hynobius species, (B) within H. kunibiki sp. nov., (C) within H. utunomiyaorum (also H. hidamontanus), (D) within H. setoi, and (E) within H. iwami, respectively. Scale bars indicate the genetic distance (expected changes per site).
Point 13
Line 459: replace “not monophyletic” by “two well-differentiated”
Response 13
We replaced it as follows: In contrast, allozyme and mitochondrial data confirm that H. utsunomiyaorum and H. kunibiki sp. nov. are two well-differentiated species (Matsui et al. 2006; Matsui et al. 2019).
Point 14
Lines 569–576: the conclusion is redundant with the introduction and adds nothing to the whole text. In my opinion, this part should be removed.
Response 14
We removed this section.
Point 15
I could not get hold of the following reference: Hayashi, M.; Ooi, K. Molecular Study on Hynobius in East Area of Shimane Prefecture, Japan: Notes on Haplotype Diversity of mtDNA and Their Distribution. Bull. Hoshizaki Green Found. 2020, 23, 99–104. Where is it possible to find it ? It is not referenced in google nor in google scholar or researchgate. Is it an article in Japanese? If so, maybe it should be relevant to mention it in the reference list.
Response 15
Yes, this article written in Japanese. In addition, the reference 7 is also written in Japanese. We added the information at the end of the sentences:
- Hayashi, M.; Ooi, K. Molecular Study on Hynobius in East Area of Shimane Prefecture, Japan: Notes on Haplotype Diversity of mtDNA and Their Distribution. Bull. Hoshizaki Green Found. 2020, 23, 99–104. (in Japanese)
- Fujita, H.; Iwata, T.; Teraoka, S. Damage example of eggs of Hynobius nebulosus (high ground type) by overfishing in Eastern Shimane Prefecture. Bull. Hoshizaki Green Found. 2016, 19, 253–255. (in Japanese)
Thank you for your time and attention.
Sincerely yours,
Hirotaka Sugawara

Reviewer 2 Report
I have read the paper and found strong data supporting the recognition of H. kunibiki as a valid new species. However, I recommend a few modifications to the manuscript to make it more interesting to a general audience.
- Why use the term Supraspecies? This a conflicting terminology. If authors want to use this term it needs to be clearly defined.
- Fig 2. Why is figure one with so many trees and they are not labeled a,b,c. etc? What is the point of all the extended trees?
- Fig. 2. Remove all text about morphological analyses. It is not informative there.
- Fig. 2. The BPP and BSV values are given in different shapes. I find this very confusing. Why not just give the number.
- I think the discussion needs to be largely modified. As it is now it is mostly results proving over and over again the differences between lineages. For a more general audience the discussion should be modified to a more general topics.
Author Response
Dear Reviewer 2,
Thank you for your critical comments.
We respond to your comments with the following.
Point 1
Why use the term Supraspecies? This a conflicting terminology. If authors want to use this term it needs to be clearly defined.
Response 1
We followed the classification by Dubois and Raffaëlli (2012). However, we cannot strictly define the term in this manuscript because morphological comparisons among these supraspecies are not performed. So, we removed the term in Fig. 2, and the term described in Line 25 and 40 have been revised to “genetic clades”. Finally, we also revised and added some sentences as follows:
Line 25
supraspecies → genetic clades
Line40
supraspecies → genetic clades
Line 542–544
According to Dubois and Raffaëlli (2012), Japanese Hynobius are divided into five supraspecies (i.e., hidamontanus, lichenatus, naevius, nebulosus, and stejnegeri);
After Line 553
Dubois and Raffaëlli (2012) used the term supraspecies, but the common morphological characteristics of each supraspecies have not yet been evaluated to the best of our knowledge; hence, this term should not be used without an initial morphological definition. If the supraspecies are valid, further morphological studies among the groups are necessary.
Point 2
Fig 2. Why is figure one with so many trees and they are not labeled a,b,c. etc? What is the point of all the extended trees?
Response 2
We labeled these figures, and explanations of each figure were added as follows (we also revised Line253–257): Five phylogenies indicate that phylogenetic relationships of (A) all Japanese Hynobius species, (B) within H. kunibiki sp. nov., (C) within H. utunomiyaorum (also H. hidamontanus), (D) within H. setoi, and (E) within H. iwami, respectively. Scale bars indicate the genetic distance (expected changes per site).
Point 3
Fig. 2. Remove all text about morphological analyses. It is not informative there.
Response 3
We removed the additional information from Fig. 2.
Point 4
Fig. 2. The BPP and BSV values are given in different shapes. I find this very confusing. Why not just give the number.
Response 4
We replaced shapes by numbers in Fig. 2.
Point 5
I think the discussion needs to be largely modified. As it is now it is mostly results proving over and over again the differences between lineages. For a more general audience the discussion should be modified to a more general topics.
Response 5
We picked up the main points and largely revised the Discussion section (we deleted the lines 413–441, 482, 487–500, 502–510, 519–532).
Also, some general topics about taxonomy of Japanese Hynobius was added in the section as follows (after Line 556):
Following this description, the number of Japanese Hynobius spp. is 38, but some taxonomic problems have been left unsolved. Matsui et al. (2019) suggested that H. akiensis and H. nebulosus are not monophyletic species, and they may be divided into two species. Further, the study only described male morphology; thus, female morphology of many Hynobius species remains unidentified. The task of clarifying the distribution ranges of lentic Hynobius also needs to be undertaken as detailed distribution ranges of H. setouchi, H. akiensis, and H. vandenburghi are still unknown. Future studies involving taxonomic reexamination and field surveys to clarify the distribution ranges of both sexes of Japanese Hynobius species are required.
Thank you for your time and attention.
Sincerely yours,
Hirotaka Sugawara
